## Technical note: A low-cost approach to monitoring relative streamflow dynamics in small, headwater streams using timelapse imagery and a deep learning model

- 5 Phillip J Goodling<sup>1</sup>, Jennifer H Fair<sup>2</sup>, Amrita Gupta<sup>3</sup>, Jeffrey D Walker<sup>4</sup>, Todd Dubreuil<sup>2</sup>, Michael Hayden<sup>2</sup>, Benjamin H Letcher<sup>2</sup>
  - <sup>1</sup>U.S. Geological Survey, Earth System Processes Division, 5522 Research Park Drive, Catonsville, Maryland, 21228
- <sup>2</sup>U.S. Geological Survey, Eastern Ecological Science Center, S. O. Conte Research Laboratory, One Migratory Way, Turners 0 Falls, MA 01376
  - <sup>3</sup> Microsoft Corporation AI For Good Lab
  - <sup>4</sup>Walker Environmental Research, LLC
- 15 *Correspondence to*: Phillip Goodling (pgoodling@usgs.gov)

Abstract. Despite their ubiquity and importance as freshwater habitat, small headwater streams are under monitored by existing stream gage networks. To address this gap, we describe a low-cost, non-contact, and low-effort method that enables organizations to monitor relative streamflow dynamics in small headwater streams. The method uses a camera to capture repeat images of the stream from a fixed position. A person then annotates pairs of images, in each case indicating which image has more apparent streamflow or indicating equal flow if no difference is discernible. A deep learning modelling framework called Streamflow Rank Estimation (SRE) is then trained on the annotated image pairs and applied to rank all images from highest to lowest apparent streamflow. From this result a relative hydrograph can be derived. We found that our modelled relative hydrograph dynamics matched the observed hydrograph dynamics well for 11 cameras at 8 streamflow sites in western Massachusetts, Higher performance was observed during the annotation period (median Kendall's Tau rank correlation 0.75 with range 0.6-0.83) than after it (median Kendall's Tau 0.59 with range 0.34 – 0.74). We found that annotation performance was generally consistent across the eleven camera sites and two individual annotators and was positively correlated with streamflow variability at a site. A scaling simulation determined that model performance improvements were limited after 1,000 annotation pairs. Our model's estimates of relative flow, while not equivalent to absolute flow, may still be useful for many applications, such as ecological modelling and calculating event-based hydrological statistics (e.g., the number of outof-bank floods). We anticipate this method will be a valuable tool to extend existing stream monitoring networks and provide new insights on dynamic headwater systems.

## 1 Introduction



Small headwater streams make up 50-70% of stream network length (Benda et al., 2004; McManamay and DeRolph, 2019) and are fundamental units of riverine networks. Streamflow dynamics in these streams are crucial controls on aquatic ecosystem function (Carlisle et al., 2017; Colvin et al., 2019; Hitt et al., 2022), thermal changes, and the routing of sediment and contaminants. Headwater streamflow dynamics are uniquely complex for the following reasons: 1) a majority of small (second-order or less) stream channels dry out seasonally or during drought events (Jaeger et al., 2021; Messager et al., 2021), 2) along-channel changes can be abrupt due to geologic controls and focused groundwater inputs (Briggs et al., 2018), and 3) due to small catchment size, these streams are particularly susceptible to drastic hydrologic alterations, both anthropogenic (damming, impervious surface runoff) and natural (ice or beaver damming, wildfire effects, geomorphic changes).

Despite their importance and vulnerability, headwater and non-perennial streams are underrepresented by streamflow monitoring networks in the United States. (Deweber et al., 2014; Seybold et al., 2023) and across the world (Krabbenhoft et al., 2022). Three primary limitations lead to a sparse headwater monitoring network: first, monitoring and maintaining traditional stage-discharge gage records (Turnipseed and Sauer, 2010) to a high quality requires expertise and training that limits the number of organizations able to collect the records. Second, velocity measurements in small, shallow, and slow-moving streams are difficult to collect and have high uncertainty, making the percentage error of streamflow discharge much higher in small streams than large streams (Horner et al., 2018; King et al., 2022; Levin et al., 2023; McMillan et al., 2012). Third, in-stream instruments to measure stage in headwater streams are frequently lost or damaged due to shifting streambeds, very high local velocities, and beaver or other animal activity. Even disregarding the challenges in collecting the data, where streams are non-perennial or form disconnected pools, traditional pressure transducer-based stage measurements provide incomplete information regarding (dis)connectedness of the stream channel, making these records inadequate for certain uses in ecohydrological modelling (Steward et al., 2012).

Streamflow monitoring using imagery is an attractive alternative to in-stream instruments and has grown in popularity as camera technology has improved. Collecting imagery is appealing because it requires very little training or specialized equipment. However, analysing a large volume of imagery can be a challenge; a range of approaches has been introduced to date. Initially, manual interpretation (Schoener, 2018) or rules-based image processing techniques (Chapman et al., 2022; Gilmore et al., 2013; Leduc et al., 2018; Noto et al., 2022) were used to automate the reading of a staff gage placed in the channel. While effective and low-cost, these staff-plate based approaches still require the installation of in-channel infrastructure that may not be permitted in protected lands or can be damaged by high flows. Additionally, stage monitoring is restricted to the location of the staff plate; therefore, any debris on the staff plate or view blockage due to snow or vegetation will result in missed readings. Computer-vision based approaches that avoid the use of an in-channel staff plate have been introduced, but generally require the manual identification of a specific region of interest in the image (Keys et al., 2016), image orthorectification using ground control points, and detailed high-resolution 3D models of riverbed and bank geometry to estimate changes in stage (Eltner et al., 2018).

Advances in deep learning approaches for imagery analysis have created new opportunities for environmental monitoring. For example, several recent studies have applied deep learning to image-based stream stage monitoring to eliminate the need for fixed in-stream staff plates. Many of these papers use established image segmentation algorithms (i.e. convolutional neural networks) to classify parts of the image as "water" or "not-water" (Eltner et al., 2021; Liu and Huang, 2024; Vandaele et al., 2021). Using a reference point on the image and knowledge of the interface location, the stream level is tracked over time. While effective, these approaches are sensitive to channel rearrangement or view blockage at the water/not-water interface. They also still require some manual judgement about the location of interest in the image frame for which stage is provided and image orthorectification using ground control points.




Unlike other deep learning approaches for streamflow estimation, Streamflow Rank Estimation (SRE) was developed to minimize the need for external monitoring data to train a model (Gupta et al., 2022). The approach aims to estimate streamflow dynamics without the need for traditional discharge observations, an in-channel staff plate, designating a region of interest, or imagery orthorectification. SRE uses a learning-to-rank framework that is trained using many pairs of stream images, with discharge in the images of each pair visually compared, removing the need for stream discharge training data. We refer to the person-generated pairwise ranks as "annotations". The model is trained using the annotations to sort images from high apparent streamflow to low apparent streamflow by fine-tuning a convolutional neural network (a ResNet-18 (He et al., 2015) architecture pretrained on ImageNet (Deng et al., 2009)) and using a learning-to-rank approach utilizing the RankNet loss function (Burges et al., 2005). The rank of each image can be used to create a streamflow percentile which is correlated with the streamflow discharge and can be interpreted as a dimensionless hydrograph. While the absolute streamflow could be estimated from the streamflow percentile using an assumed streamflow discharge distribution, for unmonitored catchments this distribution would need to be estimated independently of the SRE model and would be a significant source of uncertainty in absolute streamflow estimates (Gupta et al., 2022). As a trade-off for low-effort model training and minimal external information requirements, the rank-based streamflow percentile estimate is the primary output produced by the SRE model.

To date, the SRE model has been tested at a limited number of sites with simulated annotations derived from known streamflow discharge timeseries, but not with annotations created by people. With simulated annotations, SRE characterized streamflow percentile dynamics with a Kendall's rank correlation greater than 0.7 in five of six stream locations (Gupta et al., 2022). The number of annotations (n = 500, 1000, 2500,10000) and annotators' ranking ability (could discern 0%, 10%, 20%, 50% discharge difference) both strongly influenced the model's ranking performance. This promising early work motivated us to further evaluate the real-world performance of the model by using person-generated annotations and expanding the number of stream sites at which we assessed model performance. With a better understanding of the factors influencing model performance, we plan to apply SRE to currently unmonitored headwater catchments.

This paper describes a methodology for monitoring relative streamflow dynamics in small headwater streams using timelapse imagery coupled with a deep learning model trained using person-generated annotation. We evaluate the real-world performance of this monitoring system and answer the following questions:

- 1. How accurate are people at ranking images by streamflow?
  - 2. How accurate are the image-derived relative hydrographs developed using person-generated annotations?
  - 3. Which factors influence ranking model accuracy and can indicate which unmonitored catchments would be suitable for low-cost camera monitoring?
  - 4. How many person-generated annotations are required to achieve stable ranking model performance?

## **105 2 Methods**





## 2.1 Data Collection

To collect timelapse imagery from low-cost cameras, this project developed a web platform titled the Flow Photo Explorer (https://www.usgs.gov/apps/ecosheds/fpe/). Since its inception in October 2021, the Flow Photo Explorer (FPE) platform has accepted imagery submissions from an array of organizations with a common motivation of enhancing and expanding stream monitoring networks. While guidelines are provided on the webpage, there are few restrictions on how cameras are configured and what views they capture. The only requirement is that the imagery format uploaded to the FPE platform is formatted with EXIF metadata, which is a common imagery data format across many low-cost battery-powered game or trail cameras. We recommend a photo every 15 minutes, though the FPE database contains intervals from less than 5 minutes to once per day. The recommended camera view is looking downstream or upstream, though based on field conditions some sites may instead feature cross-stream or tangential views. We expect that the image-based monitoring approach will work best when at least some fixed objects (i.e. trees, boulders, bridge pilings, stream banks) are visible at all levels of streamflow. An example camera view with these fixed features visible is shown in Fig. 1. If a user knows a U.S. Geological Survey (USGS) stream gage monitoring the same stream reach, they can indicate the USGS station identifier and data are automatically pulled from the USGS National Water Information System (U.S. Geological Survey, 2024) database. Alternatively, they can upload their own streamflow observations, although they are not required. To test the methodology, we co-located 11 cameras with eight USGS gages in western Massachusetts for which records of stream discharge are available (Fair et al., 2025). Four cameras were located at the same streamflow monitoring location to examine the effect of differing camera angles on monitoring performance. In this study we collected imagery every 15 minutes with Reconyx (Hyperfire 2 model) Bushnell (Trophy and Essential models) cameras that were mounted to trees (except for one site that was affixed to a bridge) using swivel mounts and a secure metal housing.

Figure 1 – The recommended camera view includes stream banks and fixed objects such as trees or boulders visible at most flows. Photograph by the U.S. Geological Survey.

For this analysis we set minimum data availability criteria to test the method at sites with sufficient data. We expected that seasonal changes in vegetation, streamflow, and snow cover would appear in the imagery. Therefore, we selected sites with stream discharge and imagery data that spanned at least 1.5 years. We implemented this criterion to ensure that the model training period spanned at least one full year, so that all seasons were represented, and so that we additionally had access to a final half year of data for testing purposes. Within this span, we allowed some data gaps, since these are common in our available set of imagery data. We required at least 180 complete days of data within the 1.5 years, which is a completeness of approximately 33%. **Table 1** contains a list of sites that met our data availability requirements. These locations are mapped in **Fig. 2**. In this analysis we used daytime-only imagery (from 7 am to 7 pm), though many sites have cameras with an infrared flash that also produce usable imagery at night.

To guide user site selection and setup, we evaluated patterns in model performance according to two key site attributes. The first is a measure of flow variability during the monitoring period. Some streams, such as those heavily influenced by groundwater discharge can have small fluctuations in stream stage that are difficult to identify in imagery. We selected the coefficient of variation (CV) of log-transformed streamflow log<sub>10</sub>(Q) to quantify the general variability of the stream. The second metric is a simple qualitative assessment of how stable the camera view is over the period of record. This metric is primarily for quantifying if there were abrupt changes in the field of view of the image time series, mainly coinciding with when the camera was serviced. Cameras can also shift slightly due to vibrations or wind changing the mounting position, though these types of shifts are minor alterations compared to abrupt view changes. In this rating system, a camera stability value of "Low" indicates that there was at least one camera view change of 50% or greater (i.e. only half of the original frame was still visible). "Medium" indicates at least one camera view change between 25% and 50%, while "High" indicates that all view changes were below 25%. These two attributes were selected to inform user site selection and field methods.

| Location<br>ID | Station Name<br>(USGS Station<br>ID)     | Monitoring<br>Period        | % of images have observed stream flow | Number of<br>Annotations | Training period CV of log10(Q) | Camera<br>stability | Drainage<br>area<br>(km²) |
|----------------|------------------------------------------|-----------------------------|---------------------------------------|--------------------------|--------------------------------|---------------------|---------------------------|
| ABB            | Avery Brook<br>Bridge<br>(01171000)      | 2021-03-10 to<br>2024-04-02 | 99.1                                  | 3,147                    | 0.8                            | Low                 | 7.8                       |
| ABL            | Avery Brook<br>River Left<br>(01171000)  | 2021-07-02 to<br>2024-04-02 | 98.8                                  | 2,277                    | 0.8                            | High                | 7.8                       |
| ABR            | Avery Brook<br>River Right<br>(01171000) | 2021-03-19 to<br>2024-04-02 | 99.3                                  | 2,214                    | 0.8                            | Medium              | 7.8                       |
| ABS            | Avery Brook<br>Side<br>(01171000)        | 2021-03-19 to<br>2024-04-02 | 99.2                                  | 2,441                    | 1.0                            | High                | 7.8                       |
| GR             | Green River (01170100)                   | 2022-09-29 to<br>2024-03-29 | 99.2                                  | 5,057                    | 0.2                            | High                | 107.2                     |
| SB             | Sanderson<br>Brook<br>(01171010)         | 2021-04-01 to<br>2024-03-22 | 70.9                                  | 4,821                    | 1.2                            | Low                 | 4.4                       |
| WB0            | West Brook 0 (01171100)                  | 2022-02-01 to<br>2024-04-02 | 99.1                                  | 7,953                    | 0.8                            | High                | 27.7                      |
| WBL            | West Brook<br>Lower<br>(01171070)        | 2019-02-27 to<br>2024-04-09 | 67.7                                  | 2,256                    | 0.7                            | High                | 21.8                      |
| WBR            | West Brook<br>Reservoir<br>(01171020)    | 2021-03-25 to<br>2024-03-22 | 64.8                                  | 2,325                    | 1.1                            | High                | 16.1                      |
| WBSR           | West Branch<br>Swift River<br>(01174565) | 2017-09-14 to<br>2024-03-28 | 99.5                                  | 3,553                    | 0.3                            | Medium              | 32.6                      |
| WW             | West Whately (1171005)                   | 2021-04-06 to<br>2024-04-09 | 70.2                                  | 2,510                    | -2.5                           | Medium              | 1.3                       |

**Table 1 -** Summary of data collected at locations included in this analysis. Streamflow observations were originally reported in a U.S. Geological Survey (USGS) data release (Fair et al., 2025). "Training period CV of  $log_{10}(Q)$ " refers to the coefficient of variation of log-transformed streamflow discharge during the model training period.

Figure 2 - Map of monitoring locations in western Massachusetts, USA (Fair et al., 2025; Goodling et al., 2025). Triangles in panels C and D indicate monitoring sites and are labelled with site identifiers listed in Table 1. Arrows in Panel D indicate streamflow direction. Water bodies shown are from the NHDPlus Version 2 (McKay et al., 2012) (panel C) and NHD High Resolution (Moore et al., 2019) (panel D) datasets.

## 2.2 Data Annotation



Training the neural network model to predict streamflow dynamics from imagery requires external site-specific information. Because we hope to use this method in places with no other information except for the imagery, we could not use any streamflow data in model training. Instead, we relied on people to rank pairs of images by streamflow in a process called 'data annotation'. In the FPE web application, users were shown two photos from a given site side-by-side and asked to indicate which one had more streamflow (**Fig. 3**). The images selected to form a pair were selected at random. The users also indicated if the images appear "about the same" or if the image was a "bad photo" (obscured or too dark). "Don't know" was selected if the photo is bad or if other aspects of the images made them difficult to compare, such as a large difference in camera view or camera angle. Image pairs marked "don't know" were not used in model training. In this study, users were only presented with images collected during daytime (7am – 7pm). A typical user completed an annotation in 1-3 seconds on average; if focused, an individual could perform approximately 1000 annotations in an hour. Our dataset includes 17 unique annotators who

contributed to the model training; however, only two annotators represent 93.7% of all the annotations and we focus on these two in our discussion of annotator performance. Both of these annotators were student interns (one ecology graduate student, one environmental science undergraduate student). The student interns were associated with the project but had no specialized training or experience in streamflow monitoring.

*Figure 3* – The web-based annotation interface from the Flow Photo Explorer used in this study to develop training datasets for the ranking model.

The process of annotation was not error-free; the judgments made by individual annotators could sometimes be incorrect. This could be through simple errors of transcription (i.e. clicking the incorrect button) or because the imagery pairs were difficult to compare because of lighting, vegetation, or seasonal differences. These errors, if significant, could provide spurious information to the deep learning model. We therefore quantified the performance of our annotation dataset using the known true flow-based ranks from the co-located USGS gage data. Our primary metric was classification accuracy for the selection of the "left" or "right" image with higher streamflow in the image pair:

(1) Classification % Accuracy = 
$$\frac{TL+TR}{TL+FL+TR+FR} * 100$$

Where TL and TR refer to true left and true right selections and FL and FR refer to false true and right selections. We observed that the difficulty of the selection increases, and therefore the classification accuracy decreases, if the two photos had similar streamflow. To fully describe annotation performance, we provide our metrics as functions of the relative flow difference between the images. The relative flow difference ( $\Delta_{rel}$ ) between a pair of photos shown to an annotator was calculated as:

(2) 
$$\Delta_{rel} = \frac{|Q_1 - Q_2|}{\frac{1}{2}(Q_1 + Q_2)}$$



Where  $Q_1$  and  $Q_2$  represent streamflow values for the two images. For positive values inclusive of zero, the value  $\Delta_{rel}$  is bounded to be between zero and two. A  $\Delta_{rel}$  value near zero indicates close agreement between  $Q_1$  and  $Q_2$  whereas a  $\Delta_{rel}$  value of 2 could indicate that one of the two values is approaching zero or infinity. We compute the overall classification accuracy within binned increments of 0.1  $\Delta_{rel}$ ; the unweighted binned performance is used to develop a function describing the relationship between  $\Delta_{rel}$  and classification accuracy.

## 2.3 Modelling Methodology







Annotated images were ranked into an ordered sequence using the previously developed SRE neural network model (Gupta et al., 2022). An independent model was trained for each site. The SRE neural network model takes an image as input, which includes three channels (RGB), and generates a dimensionless, continuous-valued score representing relative streamflow as output. The score is derived by applying a sequence of mathematical operations to the input image, including spatial convolutions, which help the model extract relevant features from the image. During training, the model is given batches of paired images ranked by annotators based on relative streamflow. Two neural networks with shared model weights sequentially predict dimensionless scores for the two images. The pair of scores is used to compute a probabilistic ranking loss (Burges et al., 2005) that is minimized when the model assigns a higher score to the image that the annotator ranks as having higher flow, or assigns the same score to both images if the annotator ranks them as having the same flow. This architecture is sometimes called a "twin neural network". Images are pre-processed by resizing, centre-cropping to exclude metadata bands, and normalizing. While training, data augmentations such as random crops, horizontal flips, rotations, and colour jitter are applied to improve model robustness, generalization, and reduce overfitting (Shorten and Khoshgoftaar, 2019). Additional detail on model development and image pre-processing is available in the **supplemental materials**. After training, the model is used to generate score predictions for all images from a site, which are then standardized into z-scores by subtracting the mean and dividing by the standard deviation.

The imagery data were divided into training, testing, and validation splits to enable robust model evaluation. Unlike many machine learning applications, the model learns from image pairs and not individual images; therefore, these splits are a bit more complex to develop. When reporting model performance, we identify images that comprised pairs used for training ("train", representing 80% of annotations) or validation ("val", representing 20% of annotations). Images that were not part of any annotation pair provided to the model are used for "test". We further divided this into "test-in", which is coincident with the timeframe of annotation, and "test-out" (when available) for the period following the period with annotations. "All-in" is the combined set of images, regardless of if they are part of an annotation pair, during the annotation period. "All" is the performance for all images. We consider "test-in" to represent a retrospective model performance, while "test-out" to represent the expected performance of a deployed operational model on new imagery.

The sites in this study were co-located with traditional USGS streamflow gages, which enables us to evaluate model performance relative to these instruments. Our model performance metric is Kendall's Tau, a nonparametric rank-based correlation coefficient (Kendall, 1938). We selected this metric because it is insensitive to monotonic transformations such as

log-transformation and percentile calculations, making it appropriate to compare values on different scales and with different distributions. As a metric it is strict regarding timing; short-lived peaks, if slightly mis-timed, will result in low Kendall's Tau. Because it is based on ranks, it is insensitive to the magnitude difference between two values. As a result, low-flow observations, which are more common, have a greater influence on the resulting Kendall's Tau than short-lived high-flow observations.

To provide a preliminary understanding of the factors influencing model performance we present pairwise relationships between annotation accuracy, streamflow variability, camera stability, and model performance. For comparisons among the numeric values we present the Pearson's correlation coefficient and two-sided p-value calculated with the *cor.test* function in R version 4.3.2 (R Core Team, 2021). For comparisons between numeric values and the categorical camera stability metric, we present the results of the nonparametric Kruskal-Wallis test to evaluate if the distribution varies among the categories (Kruskal and Wallis, 1952). If significant, we perform Dunne's *post hoc* pairwise multiple comparison test to identify which categories have statistically different distributions (Dunn, 1964). The Kruskal-Wallis and Dunne's tests are computed with the rstatix R package (Kassambara, 2023).

## 2.4 Sensitivity analysis







We performed a sensitivity analysis to understand how many person-generated annotations are required to achieve acceptable performance. In this case, the target performance level was that achieved by training the model with all available image pair annotations for a given site. We created nested subsets of the annotations, beginning with increments of 100 up to 500, then using larger increments of 250 up to 1500, and finally using increments of 500 up to 3000, with additional subsets at 4000 and the maximum number of available annotations. Smaller increments were used at the lower end of the annotation range to capture the more substantial improvements in model performance that are typically observed with initial increases in training data. Each subset was a strict superset of the previous one, meaning that each larger subset contained all the pairs from the smaller subsets plus additional pairs. This allowed us to assess how increasing the volume of training data impacts model performance and to identify the point where performance plateaus, avoiding unnecessary annotation efforts that may not significantly improve performance. The sensitivity analysis reported the Kendall's Tau model performance metric is for the "test-in" data split for daytime images (7am – 7pm).

To ensure the robustness of our findings, the analysis was repeated five times. For each repetition, we randomly permuted the order of the annotations before generating the nested subsets, thereby mitigating any potential variance that could arise from the specific sequence of training samples.

## 3 Results

## 3.1 Annotation results

Annotation performance in our dataset was high (average 92.2% accuracy) and was generally consistent across sites and annotators. Accuracy was well-described by an increasing function of the relative flow difference (global 4<sup>th</sup> order polynomial, R<sup>2</sup> = 0.89, **Fig. 4**, red lines). At all sites, annotation accuracy neared 100% accuracy above a relative flow difference of 1 (which occurs when one image has three times as much streamflow as the other). As the relative flow difference neared 0, classification accuracy approached 50%, which is equivalent to guessing between the photos. Similar curves are observed for the two primary annotators (represented by symbols in **Fig. 4**). To characterize the overall accuracy of the annotation at a site, the percent accuracy of all annotations regardless of relative flow difference is reported in each panel of **Fig. 4**. The site with the lowest overall annotation performance—West Whately, with an 84% overall accuracy—had the lowest streamflow coefficient of variation a "medium" level of camera stability (**Table 1**).

# Annotation Performance Summary Only annotators with at least 1000 annotations

Adj. R<sup>2</sup>: 0.889

Figure 4 – Annotation accuracy for each site as a function of the relative difference in streamflow between the two images shown to the annotator. Percent accuracy was computed for annotations in binned intervals of 0.1 relative flow difference. Two annotators (represented with symbols and named with 5-digit alphanumeric identifier) performed annotations across the 11 camera sites. The red line is a 4<sup>th</sup> order polynomial fit across all 11 camera sites, with equation and fit statistic shown at the bottom of the figure.

## 3.2 Modelling Results



Predictions from models trained on person-generated annotations were found to represent both individual storm events and inter-annual hydrologic changes with a satisfactory degree of fidelity, with "test-in" Kendall's Tau values ranging from 0.60 to 0.83 (**Fig. 5**). We separately report statistics for the data splits "test-in", "test-out", "all-in", and "all". Most models have a slight decrease in performance (approximately 0.02) when comparing the training to test-in results. This decrease is a measure of overfit to the data. Green River has the greatest decrease (0.08, or 10%). A review of the annotations for this site shows a low density in annotations at the end of the training period that could account for this difference. Where

available, the test-out performance is lower than test-in performance (mean decrease is 0.20), suggesting a decreased ability to generalize to new flow conditions or camera views.

Within our camera monitoring dataset, we have several co-located cameras that were independently annotated and trained (lighter colour bars in **Fig. 5**). Four co-located cameras exhibited similar test-in performance, although a downstream-facing view had slightly lower performance than the other three. For the test-out period, two sites (Avery Brook River Left and Avery Brook Side) have much better performance than the other two. These sites have "high" camera stability and greater annotation accuracy than the other two sites. The streamflow has the similar (but not identical) coefficient of variation due to the differing monitoring timeframes among the cameras.






Model prediction timeseries show a clear correspondence with observed streamflow timeseries, especially when both datasets are displayed as rank percentile units (**Fig. 6**; **supplemental materials**). Major hydrologic events such as a drought that occurred in this area from June-September of 2022 and a prolonged wet period in July-August of 2023 are visible in the estimates derived solely from the imagery model. The duration and magnitude of major hydrologic events match well between observed streamflow and model predictions. Short-lived peaks from individual storm events are also well-characterized by their timing and general magnitude.

Model performance of the "test-in" set, annotation performance, flow variability, and camera stability were found to be highly interrelated (**Fig. 7**). Positive correlations were observed between flow variability and annotation accuracy (Panel A), flow variability and model performance (Panel C), annotation accuracy and model performance (Panel D). West Whately is an outlier to some extent; we report Pearson's correlation coefficients and p-values with and without this camera site. The relationship between annotation accuracy and model performance (Panel D) has the highest correlation and is least affected by the outlier presence. Camera stability, a categorical variable, was weakly related to annotation accuracy (Panel B). The Kruskal-Wallis test indicates that the annotation performance is non-identical across the three stability classes at the 0.05 significance level. The *post hoc* Dunn's pairwise multiple comparison test shows the only significant difference is between the "high" stability and "medium" stability classes. The Kruskal-Wallis test indicates there is no significant difference in Kendall's Tau among the stability classes (Panel E). Among the four cameras located on the same stream reach (shown with lighter shading), the highest performance in annotation accuracy and prediction Kendall's Tau was observed for Avery Brook River Left, which had a highly stable camera.

## Multi-site performance summary

Figure 5 – Summary of model performance, as defined by Kendall's Tau correlation, between observed and estimated streamflow percentile. Results are presented for 11 sites, 4 of which are co-located. Site abbreviations shown in brackets. Results are presented for six different sets of the data. The set "test-in" represents unseen images coincident with the training period. The set "test-out", which is not available at all locations, represents unseen images following the training period.

Figure 6 – Timeseries prediction at a single site representing intermediate model performance. Top two panels show the streamflow, middle two panels show the predicted model score, bottom two panels show both when transformed to rank percentile. The left column indicates the full period of record, the right column is an inset. In the inset plots, daily means are plotted as dots and the 15-minute interval predictions are plotted with lines. Prediction timeseries for all sites are shown in the supplemental materials.

Figure 7 – Relationships between flow variability and annotation accuracy (panel A), camera stability and annotation accuracy (panel B), flow variability and model performance (panel C), annotation accuracy and model performance (panel D), and model performance and camera stability category (panel E). Flow Variability is quantified with the coefficient of variation of log-transformed streamflow. Model performance is the "test-in" split. Point labels refer to site number listed in Table 1. The four co-located cameras are indicated with light grey square symbols. Panels A, C, and D have text indicating the Pearson's correlation coefficient and significance at the p 

fast-moving turbulent water such as the mountainous headwater streams in our dataset (Birgand et al., 2022). A deep-learning water segmentation-based approach reported Spearman correlations between independent stage measurements ranging from 0.57 to 0.94 at a single well-characterized gage site in eastern Germany (Eltner et al., 2021). We note these performance metrics reported by other similar studies, though due to differences in the model outputs our performance metrics are not directly comparable. Where evaluated in the field, most similar studies report results for single sites and/or for durations of less than 1 year (Birgand et al., 2022; Eltner et al., 2021; Leduc et al., 2018; Liu and Huang, 2024; Schoener, 2018), making this study's multi-year monitoring of 11 camera sites a comparatively robust representation of model performance.

This work, while promising, is limited in a few important ways. Primarily, this system is not (and is not intended to be) a replacement for high accuracy stream stage or discharge measurements that are required for many applications such as computing streamflow trend, calculating nutrient loads, or supporting water management decision making. Users of this system must understand the relative nature of the results and determine if relative streamflow hydrographs are suitable for their application; we envision suitable applications to include habitat characterization, aquatic species population dynamics modelling, refining process understanding in small catchment studies, intermittent stream monitoring, and characterizing event (i.e. flood or drought) timing. In this study, model training and prediction is limited to daytime imagery, which we defined simply as between 7am and 7pm local time. While these cameras also have infrared flash that illuminates the channel, the degree to which the scene is visible at night varies significantly between sites. The imagery at night becomes greyscale and we expect that different portions of the imagery become important for a model. It is unclear if nighttime imagery is best modelled with both day and nighttime imagery or if a night-only model should be trained, and future work may investigate this. We also noticed that lens fog, camera glare, vegetation blockages, and other visual impediments had a negative impact on model performance. When present, these image issues typically resulted in abrupt high or low outliers in model score. For this analysis we retained these predictions as part of the overall evaluation. We expect computer vision algorithms to detect and remove these images which would further improve model performance. Data collection on the Flow Photo Explorer platform enables users to flag "bad" images during data annotation, which will enable us to develop outlier detection algorithms for this purpose.

## 465 **5 Conclusions**






The camera-based methodology discussed here offers a novel approach to estimating relative streamflow. Its low cost and effort requirements should make it feasible to create dense observation networks to fill gaps in existing streamflow monitoring observations and thereby improve understanding of relative streamflow dynamics in headwater streams. While currently limited to estimates of relative streamflow trained as single-site models, we expect continued improvements that will expand the applicability and improve the ease of training models for new locations. The purpose of this paper was to answer questions regarding based on an initial set of monitoring stations. These findings will guide further development of the Flow

Photo Explorer integrated web platform that allows users to upload, annotate, model, and interpret headwater stream imagery. To summarize, this study answers the following questions:

- 1. How accurate are people at ranking images by streamflow? Overall annotation accuracy of image pair ranking ranged from 84% to 96% (average of 92.2%) among the 11 camera sites. While limited to primarily two individuals, we see that our annotators are nearly 100% accurate at ranking stream image pairs when there are large differences in observed streamflow. Small differences in streamflow between image pairs were more difficult for the annotators to identify. Due to consistency among sites, the accuracy of person-generated streamflow annotations used in this study can be reasonably simulated with a single globally fit equation.
  - 2. How accurate are image-derived relative hydrographs developed using person-generated annotations? Kendall's Tau values for streamflow percentile predictions ranged from 0.6 to 0.83 for unannotated days within the training period. These represent the retrospective model performance. Lower performance was observed for predictions on data collected after the training period, which may have a different distribution of streamflow or changes to the image scene. Where available, Kendall's Tau values for the post-training period range from 0.34 to 0.74.
  - 3. Which factors influence ranking model accuracy that and indicate which unmonitored catchments would be suitable for low-cost camera monitoring? The primary factor describing among-site differences in performance was streamflow variability. Describing relative streamflow changes in streams with steady flow was challenging, in part due to our relative (percentile-based) metrics of performance. We expect better performance for streams that exhibit large stage variations, are seasonally dry, or have large seasonal variations in flow.
- 4. How many person-generated annotations are required to achieve stable ranking model performance? An experiment indicated that for most sites there were diminishing improvements in performance after about 1,000 pairwise annotations. We therefore conclude this is a reasonable minimum number of annotations to develop a ranking model.

## **Code Availability**



Modelling code is provided at this GitHub code repository: <a href="https://github.com/EcoSHEDS/fpe-model">https://github.com/EcoSHEDS/fpe-model</a> (fpe-model v0.9.0).

## 500 Data Availability

The imagery, streamflow data, and model results used in this study are publicly visible on webpage (<a href="https://www.usgs.gov/apps/ecosheds/fpe">https://www.usgs.gov/apps/ecosheds/fpe</a>). Streamflow data were originally reported in a U.S. Geological Survey (USGS) data release (Fair et al., 2025). Model predictions, annotation data, and sensitivity analysis data are also available as a USGS data release (Goodling et al., 2025).

## 505 Author contribution

BL, JF, JW, AG, and PG conceptualized the study; MH and TD collected the data; JW developed the web platform; JW and AG performed modelling; PG, JW, and AG performed data analysis; PG and AG wrote the manuscript draft; PG created figures; JF, BL, and AG edited the manuscript.

## **Competing interests**

The authors declare that they have no conflict of interest.

## **Disclaimer**



This work has not been formally reviewed by the U.S. Environmental Protection Agency (US EPA). The views expressed in this document do not necessarily reflect those of the US EPA. US EPA does not endorse any products or commercial services mentioned in this publication. Any use of trade, firm, or product names is for descriptive purposes only and does not imply endorsement by the U.S. Government.

## Acknowledgements

The authors thank Josie Pilchik and Ethan Yu for their assistance with preparing the annotation datasets used in this study. We thank US Geological Survey colleagues William Farmer, Brian Pellerin, Jeffrey Baldock, and Timothy Lambert for reviews that improved the manuscript. We thank two anonymous reviewers and the journal editor (Markus Weiler) for constructive feedback that also strengthened the manuscript.

### **Financial Statement**

Funding for this project is provided through the U.S. Geological Survey (USGS) Next Generation Water Observing System (NGWOS) Research and Development Program. This article was developed in part under a Professional Services Contract (68HE0B24P0246) awarded by the U.S. Environmental Protection Agency (US EPA) to Walker Environmental Research,

LLC with funding from the US EPA Regional-ORD Applied Research (ROAR) program. Model development work was advanced through a collaborative agreement between the Microsoft Corporation AI For Good Lab and the USGS.

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
