# Peer review of "Technical note: A low-cost approach to monitoring relative streamflow dynamics in small, headwater streams using timelapse imagery and a deep learning model"

_EGUsphere, 2025_

## Author Response (AR1)

Reviewer comments in *italics*Author Responses in **bold**

**Editor comments**

Dear Authors,

as you could see from the reviewer comments, both reviewer are quite supportive of your paper, but also proposed a couple of points that need to be changes in the revised manuscript. As mentioned by both reviewer, you should clearly highlight the fact that only relative measurements are possible and I would also suggest to change the format to a Technical Note rather than a Research Article.

We made these changes in our revised manuscript and are happy to change the format to a Technical Note, which as far as we can determine from the HESS Manuscript Types webpage only involves inserting "Technical note:" at the front of the title.

In addition to the reviewer comments, I would also propose to change the format of the conclusion to a paragraph structure and not using a bullet list. This is very untypical in HESS.

We would prefer to keep the general structure of providing answers to the 4 distinct questions that we raise in the introduction section. We reformatted to minimize the use of bullets and to number the paragraphs, a structure that we also see in the conclusion section of this recently published technical note from HESS:

Lema, F., Mendoza, P. A., Vásquez, N. A., Mizukami, N., Zambrano-Bigiarini, M., and Vargas, X.: Technical note: What does the Standardized Streamflow Index actually reflect? Insights and implications for hydrological drought analysis, Hydrol. Earth Syst. Sci., 29, 1981–2002, <a href="https://doi.org/10.5194/hess-29-1981-2025">https://doi.org/10.5194/hess-29-1981-2025</a>, 2025.

I hope you can take care of the comments and provide a detailed list of changes in your response letter including reasons if you disagree with the proposed changes of the reviewers. It would also be helpful to sen a version with track changes, so we can assess the changes directly.

Best regards

Markus Weiler

We provide point-by-point responses to comments below and we will upload the tracked changes document detailing all differences made within the main text and supplemental document to respond to the reviewer comments.

Reviewer #1 comments https://doi.org/10.5194/egusphere-2025-1186-RC1 Review of "A low-cost approach to monitoring streamflow dynamics in small, headwater streams using timelapse imagery and a deep learning model"

In their paper, the authors test a low-cost method for monitoring small headwater catchments using time lapse imagery. A deep learning model is trained on pairs of images evaluated by a person; the evaluator compares apparent differences in streamflow (equal, higher, lower) for each image pair. The trained model is then applied to rank images and produce relative hydrograph.

I have read the paper with interest and find it to be innovative and well written; I think it can be published after moderate revision.

**Comments:**

Methods – Data Annotation: how are pairs of images chosen from complete set of imagery?
 How do the authors ensure that training pairs cover the full range of conditions expected for a particular site?

We select images for annotation using random sampling. While a structured or stratified sampling based on the streamflow would be a logical alternative to ensure the full range of conditions are represented, we opted to not do this because A) we would like to use this system for unmonitored sites where this stratified sampling would not be possible and B) in addition to streamflow there are other conditions (lighting, vegetation changes, camera angle changes, snow presence) that we also need to represent that would be difficult to sample in a representative way without introducing bias into our dataset.

We inserted the sentence "The images selected to form a pair were selected at random." in section 2.2.

2. Figure 6: there appears to be a large spike in predicted model score towards the end of the time series (see inset) that is not matched by a corresponding streamflow observation. Can the authors speculate as to what might be causing this discrepancy? Also, the yellow text in Figure 6 is difficult to read, consider changing color.

Yes, there are some outlier spikes at the individual 15-minute predictions at the end of the inset time period (September 13th, 2022). A review of the imagery and model results data (shown below and interactively visible here: <a href="https://www.usgs.gov/apps/ecosheds/fpe/#/explorer/14">https://www.usgs.gov/apps/ecosheds/fpe/#/explorer/14</a>) shows that rain and low light resulted in infrared camera flash being reflected back to the camera, resulting in an outlier prediction. However, the mean value for the day is only slightly affected. We changed the yellow color to green for better visibility.

Figure 1: Screenshot from Flow Photo Explorer for the Avery Brook River Right site shown in Figure 6 of the main text. The data span includes the flood event shown in the inset of Figure 6. Orange line shows 15-minute interval predictions, blue line shows observed streamflow. The orange and blue dots show the time of the image shown above the line graph. The image is showing some glare and scattering from heavy rainfall, resulting in an outlier prediction.

3. Discussion, lines 356-367: Please define "perfect annotator", "let" and "right" in this context.

We agree the sentence was a bit confusing, it now reads: "In that study, in addition to a perfect annotator that always ranked the image pair correctly, the authors simulated annotations with varying ability to discern between streamflow differences in the photo pair."

4. Discussion, line 426: can you add other examples of how relative flow data might be used? This would help to define the broader impact of this study.

The existing paragraph provided 5 example applications in the paragraph which we think is a good start. We added a mention of an application to monitoring stream intermittency.

5. Conclusions, line 468: throughout the paper, the authors are very careful to clarify that their product is relative streamflow as opposed to discharge (volume per time). It is critical to also be clear about this in the conclusions section and in the title. Otherwise, readers might see "streamflow" and conclude "volumetric flow rate". I recommend adding "relative" to line 468 as well as to the title of the paper.

We agree; added "relative" to both locations noted and in other locations throughout the paper.

6. Line 476: please define "left/right" again in this context.

We have reworded the "left/right" phrasing in the manuscript everywhere except for in direct reference to the annotation interface; here we changed "left/right selection" to "image pair ranking".

7. Can this technique be used to distinguish between the presence/absence of flow in an image? If so, the authors' method might be useful for determining the intermittency of small headwater streams, a potentially important application.

Yes! We presented on this at the American Geophysical Union conference in December 2024 – it does quite well for stream intermittency. Hopefully the topic of follow-on work. We will note this in the applications.

8. I think this paper would be strengthened by adding some discussion about how the method might be modified to transform relative flow to absolute discharge. The authors briefly mention this in the discussion (line 430), but I think this deserves some additional attention.

We don't (yet) have a proposed methodology for doing this or an understanding of the expected accuracy. So, we think this topic certainly is important but deserves its own deeper analysis and study. As we note in the sentence and the one following it, this is the topic of future work and for now we focus this paper on introducing the relative streamflow as a data product on its own.

**Reviewer #2 comments**

https://doi.org/10.5194/egusphere-2025-1186-RC2

The manuscript presents a novel and practically valuable method for the measurement of relative streamflow from camera data. This approach stands out for its accessibility to non-expert users and its relevance for applications that consider relative discharge estimations, which the authors convincingly highlight. Furthermore, the usage of the FPE web application is commendable, as it provides an important tool to improve data availability and encourage broader participation in hydrological monitoring.

The focus on relative discharge estimation is both timely and needed. However, I agree with the first reviewer that emphasis on the relative aspect should be further reinforced in the title and conclusion, to better manage reader expectations regarding the method's scope and to clarify its intended application.

We added "relative" to the title and conclusion and at a few other locations in the manuscript.

Given the clear focus on methodological development rather than novel scientific findings per se, I suggest the manuscript may be better suited as a Technical Note rather than a Research Article.

We will discuss with the editor about the best classification of the manuscript.

A few points require further clarification and elaboration:

• The current assessment of camera stability is limited to qualitative categories. While this is a good first step, more robust, quantitative approaches to evaluating movement—such as automatic image co-registration techniques (e.g., Ljubičić et al., 2021)—are available and should at least be discussed. The current three-category approach may not provide sufficient resolution to draw strong conclusions about the influence of camera motion.

We agree and we have expanded an existing sentence in the discussion noting that the classification system is rather simple. We initially explored some preliminary options (SIFT and RANSAC) for feature matching but ultimately relegated that to future work. We appreciate the reference; after reviewing the literature this is among the best and most relevant review papers of several techniques, so we decided to adopt the suggested citation.

The sentence now reads "However, the limited three-category approach in this study may limit the findings; More complex frame-tracking algorithms to quantify camera stability (i.e. (Ljubičić et al., 2021) could further improve insights into the robustness of the method to camera shifts".

While the authors emphasize the method's independence from gauge data, it is used to validate
annotator accuracy. This introduces a reliance on gauge data that should be clarified, especially
in the context of sites lacking such reference data. The authors should elaborate on how
annotator error could be quantified under such conditions, particularly for visually complex or
challenging sites.

Good point! We added the sentences: "In this study we used streamflow gage observations to quantify annotator and model performance. Where observations are not available, annotator performance could be assessed using multiple annotators assessing the same image pairs. Model performance could be evaluated using post-hoc human review using a similar approach as annotation."

Additional details on the dataset and model training process are necessary to ensure
reproducibility. For example, how many images fell into the "don't know" and other categories
(i.e., assessing potential imbalance)? Some more specifics in regard of data augmentation, i.e.,
how many training images were given after augmentation, can be provided. Furthermore, what
batch size, learning rate, scheduler type, and number of training epochs was considered? A
summary table would be a concise and helpful way to present this information.

We added some additional details on the dataset and model training to the supplemental materials section. It now includes a table as you suggested. The text and figure S1 now clarifies that all of the images used for training (80% of the annotation pairs) receive image augmentation. In the main text we also now specify the 80%/20% split for training and validation. Here is the text and table inserted into the supplemental materials describing the modelling details requested:

"Model training used the same configuration as described in (Gupta et al., 2022), with the exception of the number of epochs considered during training (raised from 15 to 20). As in that study, we used a batch size of 64, a learning rate of 0.001, a stochastic gradient descent optimizer, and a learning rate scheduler that reduces the learning rate when the validation set loss plateaus. Model training occurred on the pretrained ResNet-18 model, with the body weights frozen for the first 2 epochs and then unfrozen to allow for fine tuning for the remaining epochs. Within the 20 training epochs considered, we selected the model with the lowest validation loss as the final model. Table S1 shows the optimal number of epochs selected for each site, as well as the number of annotation pairs used for training, validation, and the proportion of annotator selections used for each site. Annotator selections of "Don't Know" were not stored or used during model development."

| Location
ID | Station Name            | Number
of
Training
Pairs | Number
of
Validation
Pairs | Optimal
Epochs
Selected
During
Training | % of Annotator Selections |       |      |
|----------------|-------------------------|-----------------------------------|-------------------------------------|-----------------------------------------------------|---------------------------|-------|------|
|                |                         |                                   |                                     |                                                     | LEFT                      | RIGHT | SAME |
| ABB            | Avery Brook Bridge      | 2512                              | 635                                 | 18                                                  | 43.0                      | 42.0  | 15.0 |
| ABL            | Avery Brook River Left  | 1817                              | 460                                 | 18                                                  | 46.1                      | 44.5  | 9.4  |
| ABR            | Avery Brook River Right | 1773                              | 441                                 | 16                                                  | 45.1                      | 46.7  | 8.2  |
| ABS            | Avery Brook Side        | 1955                              | 486                                 | 20                                                  | 40.5                      | 44.7  | 14.8 |
| GR             | Green River             | 4059                              | 995                                 | 20                                                  | 48.0                      | 45.3  | 6.7  |
| SB             | Sanderson Brook         | 3856                              | 965                                 | 20                                                  | 45.5                      | 44.9  | 9.6  |
| WBSR           | West Branch Swift River | 2838                              | 715                                 | 20                                                  | 46.2                      | 41.2  | 12.6 |
| WB0            | West Brook 0            | 6365                              | 1588                                | 19                                                  | 46.1                      | 45.3  | 8.6  |
| WBL            | West Brook Lower        | 1809                              | 447                                 | 18                                                  | 43.4                      | 49.0  | 7.6  |
| WBR            | West Brook Reservoir    | 1862                              | 463                                 | 20                                                  | 40.3                      | 45.6  | 14.0 |
| WW             | West Whately            | 2007                              | 503                                 | 19                                                  | 41.2                      | 45.7  | 13.1 |

**Table S1**: Annotation pairs used in model development and number of training epochs for each model developed in this study.

**Reference:**

Gupta, A., Chang, T., Walker, J., and Letcher, B.: Towards Continuous Streamflow Monitoring with Time-Lapse Cameras and Deep Learning, in: ACM SIGCAS/SIGCHI Conference on Computing and Sustainable Societies (COMPASS), COMPASS '22: ACM SIGCAS/SIGCHI Conference on Computing

and Sustainable Societies, Seattle WA USA, 353–363, <a href="https://doi.org/10.1145/3530190.3534805">https://doi.org/10.1145/3530190.3534805</a>, 2022.

• The current explanation of the training process could benefit from additional clarity. I am afraid that I did not fully understand the training approach. Were both images shown simultaneously to the neural network (could also a Siamese network architecture could have been considered)? Was a single model trained across all sites, or were site-specific models developed?

In this manuscript we did relegate much of the technical details of the modelling to a prior paper (Gupta and others, 2022), but we recognize an additional sentence or two could help clarify. You are correct that our architecture is sometimes called a "Siamese" network architecture. We'll clarify the architecture using an alternative phrasing as a "twin neural network":

**In section 2.3 we now say:**

"Two neural networks with shared model weights sequentially predict dimensionless scores for the two images. The pair of scores is used to compute a probabilistic ranking loss (Burges and others, 2005) that is minimized when the model assigns a higher score to the image that the annotator ranks as having higher flow, or assigns the same score to both images if the annotator ranks them as having the same flow. This architecture is sometimes called a "twin neural network".

We trained site-specific models; we inserted the sentence "An independent model was trained for each site" in section 2.3.

**References:**

Burges, C., Shaked, T., Renshaw, E., Lazier, A., Deeds, M., Hamilton, N., and Hullender, G.: Learning to rank using gradient descent, in: Proceedings of the 22nd International Conference on Machine Learning, New York, NY, USA, event-place: Bonn, Germany, 89–96, <a href="https://doi.org/10.1145/1102351.1102363">https://doi.org/10.1145/1102351.1102363</a>, 2005.

Gupta, A., Chang, T., Walker, J., and Letcher, B.: Towards Continuous Streamflow Monitoring with Time-Lapse Cameras and Deep Learning, in: ACM SIGCAS/SIGCHI Conference on Computing and Sustainable Societies (COMPASS), COMPASS '22: ACM SIGCAS/SIGCHI Conference on Computing and Sustainable Societies, Seattle WA USA, 353–363, <a href="https://doi.org/10.1145/3530190.3534805">https://doi.org/10.1145/3530190.3534805</a>, 2022.

• It would be useful to discuss a bit more how the model might perform under extreme, previously unseen conditions (e.g., large floods). I would assume failure in unseen situations - as also the author's test-out results in the supplement reveal a lot weaker performance. In the case of, e.g., usage of water segmentation the focus would be on a specific object, which might be easier to re-identify also during extreme events as appearance of water does not change as strongly. A

discussion on potential failure modes in such situations compared to more object-specific segmentation methods could be informative.

We consider the "test-out" split performance to represent performance in unseen situations; as we note in the paper there is a decrease in model performance. We are a little puzzled about what one would call "failure" in this setting, but we expect an out-of-distribution large flood event might be identifiable in the ranking model predictions but may have an incorrect magnitude. A perhaps greater risk, which we noted in lines 398-400 is changes to the scene not observed during annotation that could manifest in over- or under-prediction.

We don't feel that our work would support much comparison to object-specific segmentation methods (especially for out-of-distribution flows in the `test-out` period) without conjecture; we limited this paper's scope to comparing our results reported performance statistics for studies using these kinds of segmentation approaches. A designed future study would be needed to evaluate the performance using the same imagery data.

To respond to the question of out-of-distribution flows in the 'test-out' period, in line 400 we inserted the clause "The general approach we took may be limited in its ability to describe the magnitude of out-of-distribution streamflow in the `test-out` period, but...".

• The authors touch on some environmental factors like fog and glare. These, along with perspective distortions (e.g., using Gaussian Splatting if multi-view images are available), could be more deeply addressed using advanced augmentation strategies in a future study. This may offer paths toward increasing the robustness of the method under diverse conditions.

We appreciate the suggested directions. We don't anticipate having multi-view images available for our target simple case of using single trail camera in a fixed position for monitoring, so Gaussian Splatting wouldn't be an option here unfortunately.

Overall, this manuscript introduces an innovative and accessible tool for relative streamflow estimation, with clear real-world utility that can benefit the broader hydrological community. While the methodological foundation is solid, the manuscript could be strengthened by clarifying certain technical aspects. I recommend publication after minor revisions and strongly support reformatting the submission as a Technical Note.

**Minor Comments:**

Line 120–122: When referencing "nearby" gauges, please specify the distances involved and potential hydrological complexities (e.g., potential tributaries in-between) that might impact the comparability of measurements.

You are correct that we should be more specific than "nearby USGS stream gage" – we replaced with "USGS stream gage monitoring the same stream reach".

Line 275–279: This section includes some repetition and can be shortened or even removed.

**We edited this section to reduce redundancy with an earlier section that defines the data splits.**

Reference (Please, do not see my listed reference as a request to be added to your references, but solely as a suggestion for more information!):

Ljubičić, R., Strelnikova, D., Perks, M.T., Eltner, A., Peña-Haro, S., Pizarro, A., Dal Sasso, S.F., Scherling, U., Vuono, P., Manfreda, S. (2021): A comparison of tools and techniques for stabilising unmanned aerial system (UAS) imagery for surface flow observations. Hydrology and Earth System Sciences, 25, 5105–5132